# MAG-Edit: Localized Image Editing in Complex Scenarios via Mask-Based Attention-Adjusted Guidance

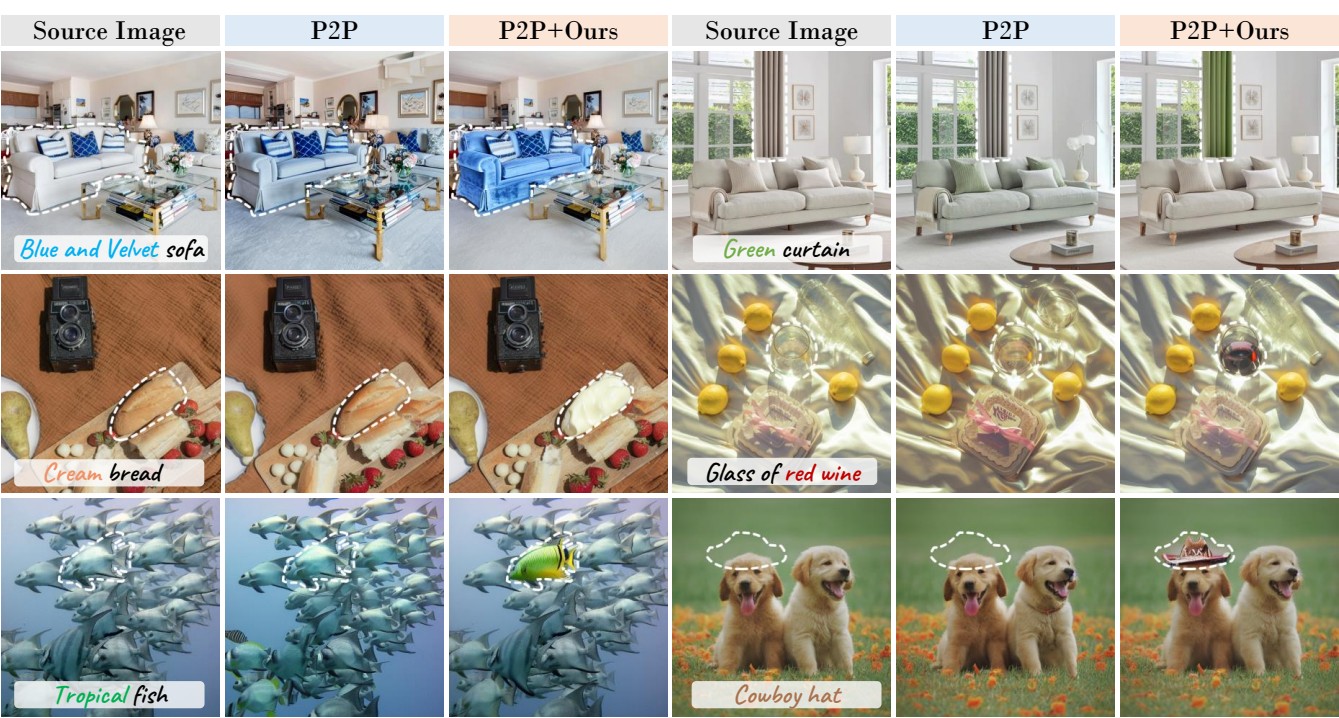

**Figure 1: Localized image editing in complex scenarios. The editing regions are demarcated by white dashed lines, with the target edit tokens emphasized on each source image for clarity. Existing attention-based image editing methods, such as Prompt-to-Prompt (P2P) [8], face challenges in achieving precise alignment of local editing areas with text prompts in complex scenarios that involve multiple objects. This often results in the editing effects inadvertently extending beyond the intended area, impacting incorrect regions (1st row), or producing minimal effects on the target region (2nd and 3rd rows). In contrast, our method MAG-Edit, introduces a plug-and-play, inference-stage optimization strategy to enhance the capabilities of attention-based editing baselines, *e.g.*, P2P [8], facilitating accurate and effective localized edits in intricate compositions.**

## ABSTRACT

Recent diffusion-based image editing approaches have exhibited impressive editing capabilities in images with one dominant object in simple compositions. However, localized editing in images containing multiple objects and intricate compositions has not been well-studied in the literature, despite its growing real-world demands. Existing mask-based inpainting methods fall short of retaining the underlying structure within the edit region, causing noticeable discordance with their complex surroundings. Meanwhile, attention-based methods such as Prompt-to-Prompt (P2P) often exhibit editing leakage and misalignment in more complex compositions. In this work, we propose **MAG-Edit**, a plug-and-play, inference-stage optimization method, that empowers attention-based editing approaches, such as P2P, to enhance localized image editing in intricate scenarios. In particular, MAG-Edit optimizes the noise latent feature by encouraging two mask-based cross-attention ratios of the edit token, which in turn gradually enhances the local alignment with the desired prompt. Extensive quantitative and qualitative experiments demonstrate the effectiveness of our method in achieving both text alignment and structure preservation for localized editing within complex scenarios.

**Unpublished working draft. Not for distribution.**

## CCS CONCEPTS

• **Computing methodologies** → **Computer vision**.

# KEYWORDS

Text-based image editing, Diffusion models, Gradient guidance

## 1 INTRODUCTION

Recent advancements in large-scale text-to-image (T2I) diffusion models [18, 20, 22] have demonstrated their remarkable ability to generate high-quality and diverse images that reflect specified textual descriptions. Trained on extensive datasets, these models effectively link textual descriptions with corresponding images, opening up new possibilities for text-based image editing. The past year has witnessed a substantial increase in the development of methods using diffusion models for text-based image editing, which can be broadly categorized into three main categories: instruction-based training [3, 28], fine-tuning [11, 21, 29], and training-free methods [4, 7, 8, 14, 15, 25]. In this work, our focus is on the exploration and enhancement of training-free editing approaches.

Existing training-free approaches predominantly concentrate on manipulating *prominent* objects within *simple* compositions, as demonstrated in Fig. 2, where text prompts effectively identify the intended editing area. However, real-world images often feature *complex* compositions with *multiple* objects, posing challenges for precise regional edits. For instance, in home interior design, users may want to alter the color and texture of specific furniture pieces, like modifying the color of two pillows to better align their aesthetic with the overall design theme. Relying solely on text prompts struggles to accurately pinpoint the desired area in such intricate settings. While existing mask-based inpainting techniques [1, 2, 7, 9, 26], *e.g.*, Blended Latent Diffusion (Blended LD) [1] generate and integrate new objects into images under the guidance of local masks, this line of approaches leads to significant structural alterations within edited regions, disrupting the visual harmony with complex backgrounds, as shown in Fig. 2.

Balancing fidelity and editability in localized regions within complex scenarios remains a considerable challenge. Attention-based editing methods, such as Prompt-to-Prompt (P2P), can maintain the original image's structure and layout. However, their effectiveness is compromised in complex scenarios due to their reliance on the text prompts' ability to localize editing regions. Consequently, edits may extend beyond the intended areas, affecting incorrect regions, as illustrated in Fig. 2. Although integrating mask-based blending operations into P2P can mitigate unintended edits outside targeted areas, challenges persist in achieving precise alignment with the intended text prompts. Such misalignment results in edits failing to appear in the desired locations.

To address the aforementioned challenges, we propose a novel inference-stage optimization strategy, termed *Mask-Based **A**ttention-**A**djusted **G**uidance* (**MAG**-Edit). This plug-and-play approach is designed to augment attention-based methods such as P2P for more accurate localized image editing in complex scenarios, eliminating the necessity for additional training. Given that cross-attention (CA) maps in pre-trained T2I diffusion models effectively capture the correlation between input features and text embeddings, our key insight is that *adjusting the noise latent feature to attain higher CA values significantly enhances its alignment with the corresponding text prompt*. Therefore, we propose locally optimizing the noise latent feature during the inference stage using two distinct mask-based

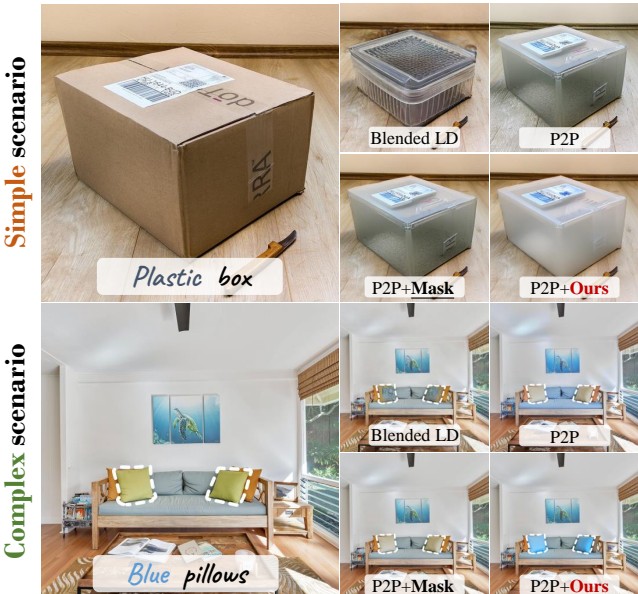

**Figure 2: Localized editing comparisons between *Simple* and *Complex* scenarios. Mask-based inpainting methods like Blended LD [1] often introduce structural inconsistencies, leading to noticeable discrepancies with the surrounding context. Due to the reliance on the text prompts' localization capabilities, P2P [8] excels in localized editing on the dominant object but struggles to precisely pinpoint the local editing region in intricate scenarios. Integrating mask-based blending techniques into P2P (*i.e.*, P2P+Mask) can alleviate leakage, but the misalignment leads to the absence of editing effects in the intended areas. In contrast, our MAG-Edit enhances P2P's localized editing performance with superior structural preservation in complex scenarios.**

CA constraints tailored for the target editing prompt. In particular, our approach aims to maximize two aspects of ratios: first, the CA value of the edit token in relation to all token CA values within the masked area, and second, the CA value of the edit token inside the mask compared to its overall CA values. Subsequently, the gradients of these constraints guide the update of the noise latent feature, thus progressively aligning the editing effect with the desired text prompt and spatial requirements. The effectiveness of the proposed method is evident in Fig. 1 and Fig. 2.

Our contributions can be summarized as follows:

- We introduce MAG-Edit, a novel plug-and-play, inference-stage optimization scheme, marking a pioneering effort to enhance existing attention-based editing frameworks for specifically addressing localized image editing in complex scenes.
- We propose two mask-based CA constraints in terms of the token and spatial ratio, guiding the local noise latent feature to better align with the target edit tokens.
- We extensively validate MAG-Edit's efficiency in localized image editing across diverse intricate scenarios. Quantitative and qualitative experimental results demonstrate a significantly improved trade-off between editing efficiency and structure preservation when compared to existing state-of-the-art approaches.

## 2 RELATED WORK

**Text-Based Image Editing Using Diffusion Models** can be mainly classified into three categories: instruction-based training [3, 28], fine-tuning [11, 21, 29], and training-free methods [4, 7, 8, 14, 15, 25]. Instruction-based training methods, such as InstructPix2Pix [3], require substantial resources for extensive training. On the other hand, fine-tuning methods, like Imagic [11], run the risk of overfitting by optimizing the full model with limited data. In this work, we specifically focus on training-free methods. One line of training-free approaches [1, 2, 9, 26] utilizes masks to generate foreground objects and blends them into the original image through blending operations. For instance, Blended Diffusion [2] and Blended LD [1], directly generate foreground objects based on text prompts. DiffEdit [7, 26] introduces an unsupervised method for learning the mask and employs DDIM inversion [23] noise latent features alongside the target prompt to generate the foreground image. Although these approaches successfully maintain the integrity of unedited regions outside of the mask, they may introduce large structural changes within the edit regions, causing inconsistencies with the surrounding context in complex scenes. Attention-based methods [4, 8, 25] such as P2P [8] involve attention integration mechanisms to maintain the structure and layout of the original image. Recent advancements in inversion methods [12–14] propose to improve DDIM inversion [23] for encoding real images, achieving improved reconstruction and more flexible editing capabilities. However, the integration of editing methods such as P2P [8] remains essential for these methods to facilitate image editing. When applied to localized editing in intricate scenarios, attention-based methods often result in leakage to incorrect areas, leading to inefficiencies in prospective regions.

**Optimization on the Noise Latent Feature.** Recent advances [5, 6, 16, 19, 27] in image generation with diffusion models have investigated the use of CA constraints to optimize the noise latent feature during inference. The pioneering work, Attend-and-Excite [5], addresses issues like catastrophic neglect and incorrect attribute binding by maximizing the largest CA units corresponding to all subject tokens in the text prompt. This approach refines the noise latent feature at each diffusion step, thereby guiding the model to generate all subjects described in the text accurately. Several training-free layout-generation methods [6, 16, 27] propose to optimize the noise latent feature by maximizing CA constraints in conjunction with bounding boxes, allowing objects to appear in specific regions. While the image generation process has demonstrated effectiveness, the application of noise latent feature optimization to image editing has received relatively less attention. Pix2pix-zero [15] offers a solution by optimizing the noise latent feature, constraining the CA maps of the editing branch to align with the reconstruction branch, thus preserving the original image's structural layout. In contrast to structural preservation, the proposed method aims to align the local noise latent feature more semantically with the target text prompt, enabling localized editing in complex scenarios.

## 3 METHODOLOGIES OF MAG-EDIT

We employ the widely used attention-based method, P2P [8], as the backbone of our approach. Preliminaries of P2P [8] are reviewed in Sec. 3.1. Importantly, our MAG-Edit technique is not confined to use with P2P [8] alone; it can also be integrated with other editing

---

**Algorithm 1:** A Denoising Step Using MAG-Edit on P2P [8]

**Input:** An original and edited prompt $\mathcal{P}$, $\mathcal{P}^*$; a timestep $t$ and corresponding noise latent features of reconstruction and editing branches $z_t$, $z_t^*$; a maximum iteration step MAX_IT; a function $\mathbf{F}(\cdot)$ for computing proposed constraint $\mathcal{L}$; a pre-trained Stable Diffusion model $SD$.

**Output:** the noisy latent feature $z_{t-1}^*$ for the next timestep of the editing branch.

1   **for** $i = 1$ **to** MAX_IT **do**
2     $\_, A_t \leftarrow SD(z_t, \mathcal{P}, t)$ ;
3     $\_, A_t^* \leftarrow SD(z_t^*, \mathcal{P}^*, t)$ ;
4     $\hat{A}_t \leftarrow Inject(A_t, A_t^*)$;     // Operation in P2P.
     $\mathcal{L} \leftarrow \mathbf{F}(\hat{A}_t)$;     // **MAG-Edit.**
5     $z_t^* = z_t^* - \mathcal{M} \odot \delta \nabla_{z_t^*} \mathcal{L}$;
6   **end**
7   $\_, A_t \leftarrow SD(z_t, \mathcal{P}, t)$ ;
8   $\_, A_t^* \leftarrow SD(z_t^*, \mathcal{P}^*, t)$ ;
9   $\hat{A}_t \leftarrow Inject(A_t, A_t^*)$;     // Operation in P2P.
10   $z_{t-1}^* \leftarrow SD(z_t^*, \mathcal{P}^*, t)\{\hat{A}_t\}$ ;
11 **Return** $z_{t-1}^*$

---

methods, such as Plug-and-Play (PnP) [25]. For further details and results, please refer to the *supplementary material* (**SM**).

Let $\mathcal{I}$ be a real image, we first encode it into the noise latent feature $z_T$ using the inversion method such as Null-text inversion [14]. Given the original text prompt $\mathcal{P}$ and edited prompt $\mathcal{P}^*$, we define the set of new target tokens as $\mathcal{S}^* = \{s_1^*, s_i^*, ..., s_I^*\}$ present in $\mathcal{P}^*$ against $\mathcal{P}$, the common tokens as $\mathcal{S} = \{s_1, s_j, ..., s_J\}$ and $\mathcal{S}^* \cap \mathcal{S} = \emptyset$. As illustrated in Fig. 3(a), the editing framework of P2P consists of two branches, *i.e.*, reconstruction and editing branches generated by prompt $\mathcal{P}$ and $\mathcal{P}^*$, respectively. Furthermore, an edit region mask $\mathcal{M}$, derived from $\mathcal{I}$, is utilized to accurately localize the editing area in complex scenarios. In this work, we aim to optimize the noise latent feature $z_t^*$ of the editing branch at the diffusion step $t$ by aligning the desired editing effects specified by $\mathcal{S}^*$ with the prospective region defined by $\mathcal{M}$, which enables localized editing in complex scenarios. To achieve this, we introduce MAG-Edit, a plug-and-play optimization strategy to automatically manipulate $z_t^*$, which contains two key steps: defining two mask-based constraints in Sec. 3.2 and performing gradient guidance in Sec. 3.3. These steps are illustrated in Fig. 3(b) and Fig. 3(c), respectively.

### 3.1 Preliminaries of Prompt-to-Prompt

P2P [8] introduces several prompt-based editing operations leveraging CA maps injection: First, the *word swap* involves injecting all attention maps in the reconstruction branch, generated by the source prompt, into attention maps from the editing process using the target prompt. In contrast, the *prompt refinement* selectively replaces the CA maps associated with tokens common to both the source and target prompts. Furthermore, P2P introduces the *attention re-weighting* operation, enabling direct scale adjustments to the CA values. This

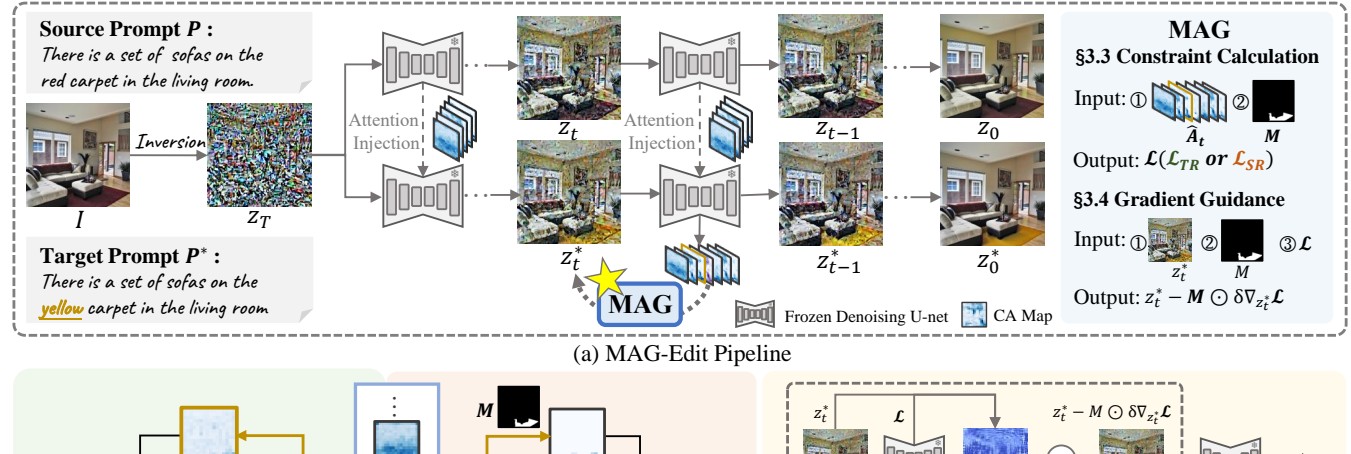

(a) MAG-Edit Pipeline

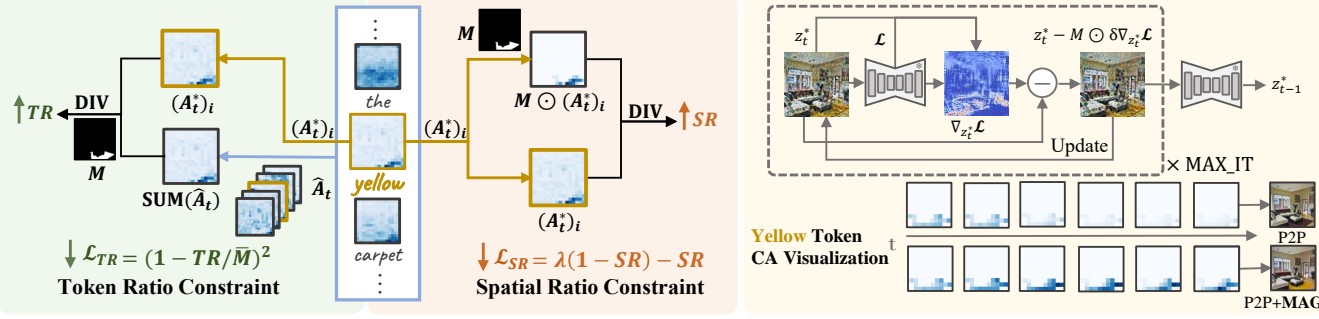

(b) Calculating Constraints

(c) Performing Guidance

**Figure 3: Illustration of MAG-Edit. (a) We apply MAG-Edit to the noise latent feature $z_t^*$ in the editing branch of P2P [8], which contains two main steps: Constraint Calculation (Sec. 3.2) and Gradient Guidance (Sec. 3.3). (b) We aim to maximize the CA values of the target token (_e.g._, "yellow") within the masked area $\mathcal{M}$ by focusing on two aspects: the token ratio (TR) and spatial ratio (SR). This is achieved by minimizing their corresponding constraints, $\mathcal{L}_{TR}$ and $\mathcal{L}_{SR}$, respectively. (c) We perform the gradient $\nabla_{z_t^*}\mathcal{L}$ as guidance to iteratively update the noise latent feature $z_t^*$ until the maximum iteration number (MAX_IT) is reached. Consequently, optimizing $z_t^*$ with MAG-Edit leads to a significant improvement in the CA values of "yellow" token within the edit regions $\mathcal{M}$, confirming enhanced alignment between the local edit region and the target prompt.**

technique is specifically designed to control the granularity of the editing process.

## 3.2 Mask-Based Attention-Adjusted Constraint

Considering that CA maps define the similarity between the input features and text embeddings, larger CA values indicate better alignment. This observation inspires the formulation of two mask-based constraints, aiming to maximize the CA value ratio in both token and spatial aspects within the predefined editing region. To illustrate, first consider the CA map $(A_t^*)_i$ of a new editing token $s_i^*$ within a specific mask region $\mathcal{M}$ such as "yellow" in Fig. 3(a). Take _prompt refinement_ operation of P2P as an example, the attention maps of common tokens, denoted as $A_t$, from the reconstruction branch are first injected into the editing process, obtaining the CA maps $\hat{A}_t$.

**Token Ratio Constraint.** Given that CA values are calculated across the dimensions of all tokens, we propose a token ratio constraint. This constraint aims to increase the proportion of the target token relative to all other tokens within the mask $\mathcal{M}$, thereby concentrating the editing effects more precisely on the designated area. As demonstrated in the left block of Fig. 3(b), the token ratio constraint is defined as follows:

$$\mathcal{L}_{TR} = \left(1 - \frac{1}{\overline{\mathcal{M}}}\sum \mathcal{M} \odot \frac{(A_t^*)_i}{(A_t^*)_i + \sum_{j=1}^{J}(A_t)_j}\right)^2, \quad (1)$$

where $\overline{\mathcal{M}}$ represents the total number of elements within the mask.

**Spatial Ratio Constraint.** In scenarios demanding significant editing granularity, the token ratio constraint might not sufficiently amplify the CA value $(A_t^*)_i$ within $\mathcal{M}$. To address this limitation, we introduce an additional spatial formulation, which is designed to maximize the CA values within the masked region while simultaneously minimizing them outside the mask, illustrated in the right block of Fig. 3(b) as,

$$\mathcal{L}_{SR} = \lambda \underbrace{\left(1 - \frac{\sum \mathcal{M} \odot (A_t^*)_i}{\sum (A_t^*)_i}\right)}_{\text{Out-mask}} - \underbrace{\frac{\sum \mathcal{M} \odot (A_t^*)_i}{\sum (A_t^*)_i}}_{\text{In-mask}}, \quad (2)$$

where $\lambda$ is a balance weight, and we set $\lambda = 3$ empirically.

**Negative Prompt Constraint.** Our proposed method can also be used to _attenuate_ the textural information associated with the original image $\mathcal{I}$ by employing negative prompts. Specifically, we define a set of negative tokens $\mathcal{S}_{\text{ng}}^*$ to represent the texture of $\mathcal{I}$ in contrast to the new tokens $\mathcal{S}^*$. Consequently, we can establish the negative prompt constraint $\mathcal{L}_{\text{ng}}$ using the negative token's corresponding CA value and optimize the noise latent feature in _the opposite direction_ as follows,

$$\mathcal{L}_{\text{total}} = \lambda_{\text{p}}\mathcal{L} - \lambda_{\text{ng}}\mathcal{L}_{\text{ng}}, \quad (3)$$

where $\lambda_{\text{p}}$ and $\lambda_{\text{ng}}$ aim to balance between positive and negative prompt constraint. For more details, please refer to the **SM**.

**Figure 4: Qualitative comparisons of localized image editing across various complex scenarios. Editing regions are highlighted with dashed lines, and simplified target prompts are shown on the source images. We compare our MAG-Edit (h) with other state-of-the-art methods including mask-based inpainting methods like (a) Blended LD [2] and (b) DiffEdit [7], attention-based methods like (c) PnP [25], (d) MasaCtrl [4] and (e) P2P [8]. We also apply the same mask for blending operation in (f) P2P+Mask and re-weight the CA values of the target token tenfold within the masked region, termed as (g) P2P+Re-weight. Our proposed method (h) not only achieves superior editing effects but also better preserves the structure in local regions against other baselines (a-g).**

| Dataset | MAG-Bench | | PIE-Bench [10] | | TEd-Bench [11] | |
|---|---|---|---|---|---|---|
| Method | CLIP Score (↑) | DINO-ViT Distance (↓) | CLIP Score (↑) | DINO-ViT Distance(↓) | CLIP Score (↑) | DINO-ViT Distance(↓) |
| Blended LD [1] | 19.12 | 0.089 | 22.58 | 0.079 | 20.56 | 0.110 |
| Diffedit [7] | 19.20 | 0.083 | 21.20 | 0.070 | 20.82 | 0.100 |
| MasaCtrl [25] | 19.00 | 0.088 | 20.44 | 0.076 | 19.08 | 0.090 |
| PnP [25] | 19.90 | 0.083 | 20.47 | 0.067 | 20.60 | 0.096 |
| P2P [8] | 20.02 | **0.079** | 21.43 | 0.074 | 20.35 | 0.091 |
| P2P+Mask | 19.77 | 0.081 | 20.95 | **0.059** | 20.25 | 0.093 |
| P2P+Re-weight | 20.62 | 0.085 | 22.15 | 0.062 | 20.10 | 0.086 |
| **P2P+Ours** | **21.79** | 0.081 | **22.76** | 0.073 | **21.36** | **0.062** |

**Table 1: Quantitative comparisons of localized image editing across diverse benchmarks. Bold and underline indicate the best and second best value, respectively.**

## 3.3 Performing Gradient Guidance

Upon establishing the mask-based constraints $\mathcal{L}$ ($\mathcal{L}_{TR}$ or $\mathcal{L}_{SR}$), we compute their gradients to determine the optimal direction for modifying the current noise latent feature $z_t^*$. In particular, to restrict the editing effect to the predefined region, we update the noise latent feature $z_t^*$ inside the mask $\mathcal{M}$ using the following equation:

$$z_t^* = z_t^* - \mathcal{M} \odot \delta \nabla_{z_t^*} \mathcal{L}, \qquad (4)$$

where the term $\delta$ represents the gradient update scale. As shown in Fig. 3(c) and detailed in Algorithm. 1, $z_t^*$ is iteratively refined until reaching the maximum number of iterations (MAX_IT).

Moreover, our proposed method can be readily adapted for multiple prompts editing as:

$$z_t^* = z_t^* - \mathcal{M} \odot \delta \nabla_{z_t^*} \sum_{i=1}^{I} (\lambda_1 \mathcal{L}^1 + \cdots + \lambda_i \mathcal{L}^i + \lambda_I \mathcal{L}^I), \qquad (5)$$

where the term $\lambda_*$ controls the editing granularity of each prompt, with their sum equaling 1. Fig. 9(a) demonstrates how our proposed method effectively balances the editing granularity for various prompts.

## 4 EXPERIMENTS

### 4.1 Implementation details

Following P2P [8], we adopt the pre-trained Stable Diffusion v1.4 [20] model as the backbone and apply Null-text Inversion [14] to obtain the noise latent feature $z_T$. All CA values are calculated in the resolution of $16 \times 16$ of the U-Net, which is known to process the most semantically rich information [5]. In our experiments, we choose MAX_IT= 15 and apply MAG-Edit during the diffusion steps $t = [T, \tau]$, where $\tau$ is specifically set to 25. The corresponding editing masks for each image are derived using the Segment Anything method[1]. To preserve the original information in the regions outside the mask, we employ the latent blending operation in P2P [8] with the derived masks instead of automatically generated ones. In practice, we select between $\mathcal{L}_{TR}$ and $\mathcal{L}_{SR}$, depending on the required granularity of the edit, allowing for adaptability across various editing types. All experiments are conducted on a single NVIDIA A100 GPU. Further implementation details and additional results using various inversion and editing baselines are available in the **SM**.

### 4.2 Comparisons with Baselines

**Benchmark Dataset.** Existing datasets for text-based image editing methods primarily focused on relatively simple scenes dominated by

[1] https://github.com/facebookresearch/segment-anything

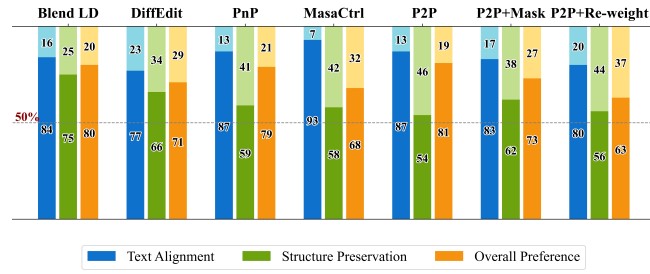

**Figure 5: Human preferences in the localized editing regions. The values presented reflect the proportion of users who favor our proposed method over comparative approaches.**

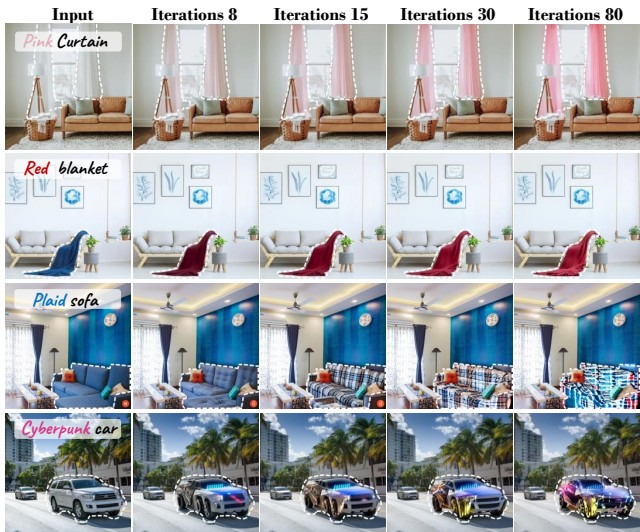

**Figure 6: Impact of optimization iterations. Increasing the number of iterations enhances the granularity of editing. However, overly extensive iterations can lead to notable artifacts arising from structural modifications.**

prominent objects. To facilitate a comprehensive evaluation of our method, we have curated a benchmark dataset named MAG-Bench, which comprises 200 images. These images showcase diverse scenes with multiple objects. Additional details of our collected benchmark are provided in **SM**. Moreover, we also assess the performance of our method on two well-used benchmark datasets *i.e.*, PIE-Bench [10] and TEd-Bench [11]. Our evaluation focuses on editing color, texture, and object replacement in localized regions.

**Compared Methods.** We conduct comparisons with existing representative training-free diffusion-based image editing methods and establish two straightforward baselines using P2P [8]:

- **State-of-the-art training-free methods**: Mask-based inpainting approaches: Blended LD [1] and DiffEdit [7]; Attention-based editing: MasaCtrl [4], PnP [25] and P2P [8].
- **Straightforward baselines on P2P**: We first combine the same blending operations used in our approach with P2P [8], denoted as P2P+Mask. Additionally, we incorporate the *reweighting* operation from P2P [8] to amplify the CA values of the target edit token tenfold within the masked area. This

enhancement aims to improve the granularity of regional editing, and is referred to as P2P+Re-weight.

It is crucial to emphasize that all mask-based methods, Blended LD [1] and DiffEdit [7], as well as two straightforward baselines, P2P+Mask and P2P+Re-weight, utilize **the same masks as our proposed method for fair comparisons.**

**Qualitative Results.** Fig. 4 clearly shows that Blended LD [1] leads to considerable structural changes in complex scenarios, resulting in significant discordance with the surrounding context. Meanwhile, DiffEdit [7], either alters the structure, as seen in the "orange table" example or fails to produce a noticeable editing effect in the intended region, as is apparent in other images. MasaCtrl [4] demonstrates limited performance in color, texture, and shape editing, resulting in negligible editing effects. Furthermore, it may introduce structural variations as presented by the "orange table" example. PnP [25] and P2P [8] often suffer leakage in adjacent regions, causing minimal effects in the prospective region. This issue is particularly noticeable in tasks such as changing the color of a red pillow to green. Although blending operations using masks in P2P+Mask can mitigate leakage in some scenarios, the misalignment issue persists, leading to inefficiencies within the intended editing regions. Even when we directly increase the CA values of the target edit token within the masked region, the resulting edits are either minimal or disrupt the structural consistency. In contrast, our proposed method exhibits improved editing performance while maintaining better structural integrity in local regions.

**Quantitative Results.** We quantitatively evaluate our proposed method against baseline models using both automatic metrics and human evaluations.

*Automatic Metrics.* To better evaluate localized editing ability, we use the bounding boxes to crop the editing regions [9] and evaluate the image-text alignment and structure preservation using the CLIP score [17] and the DINO-ViT self-similarity distance [24], respectively. Table. 1 illustrates that our proposed method significantly enhances text alignment within local regions in both simple and complex scenarios, achieving much higher local CLIP values without compromising fidelity.

*User study.* We perform a user preference evaluation via pairwise comparisons on Amazon MTurk[2], focusing on text alignment, structure preservation, and overall preference in localized editing regions. Fig. 5 indicates a notable preference among participants for our proposed method over the baselines across three aspects. A significant majority, ranging from 77% to 93%, believe that our method achieves much better text alignment compared to other methods. Furthermore, our method is preferred for better structure preservation by 75% of users over Blended LD [1]. Due to its more effective balance between editability and fidelity, our proposed method is overall favored by 68% to 81% of the participants.

## 4.3 Ablation Study

**Impact of Optimization Iterations.** The number of maximum iterations for optimizing the noise latent feature is crucial in modulating the magnitude of editing. As shown in Fig. 6, increasing the number of iterations can improve the granularity of the editing. However, in

---

[2]https://www.mturk.com/

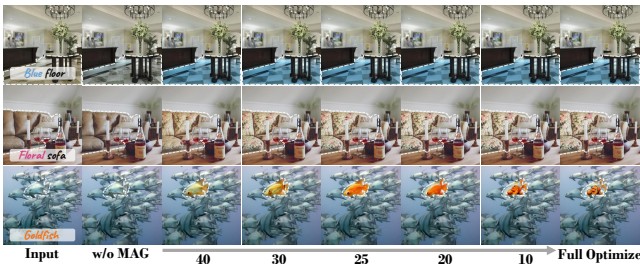

**Figure 7: Applying MAG-Edit through a varied number of diffusion steps. We use white dashed lines to demarcate the editing regions in the source images. Each row demonstrates the optimization of the noise latent feature ranging from** $0\%$ **(left) to** $100\%$ **(right) of the steps.**

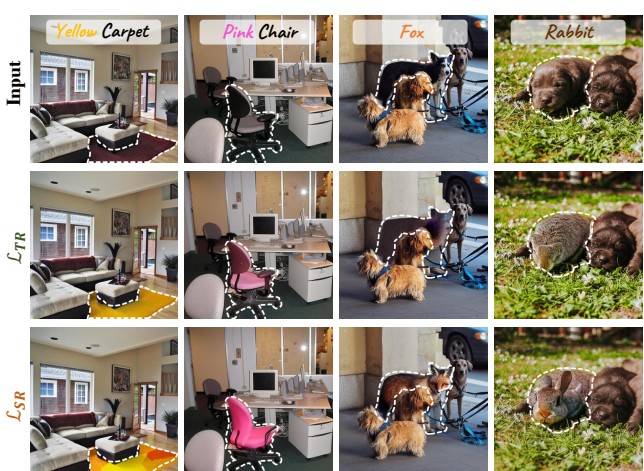

**Figure 8: Editing granularity of proposed constraints. The token ratio constraint** $\mathcal{L}_{TR}$ **efficiently preserves the inherent structure in the edited region, while the spatial ratio constraint** $\mathcal{L}_{SR}$ **enhances editing granularity.**

texture and shape editing, excessive iterations may result in significant artifacts as a result of alterations in the structure.

**Impact of Optimization Diffusion Steps.** To investigate the impact of the optimization diffusion steps using MAG-Edit, we conduct experiments with different values assigned to $\tau = \{50, 40, 30, 25, 20, 10, 0\}$, representing the end of the diffusion step range from 50 to $\tau$. As illustrated in Fig. 7, We observed that optimization within the initial diffusion steps can rapidly alter the color, indicating that optimization within the $t \in [T, 40]$ steps is generally sufficient for achieving effective color editing. In contrast, texture and shape edits require a greater number of diffusion steps. Overall, updating the latent noise feature after 25 steps does not significantly improve the granularity of texture editing but prolongs the optimization time. For shape editing, excessive optimization beyond 25 steps can lead to pronounced artifacts due to structural changes.

**Impact of Proposed Constraints.** $\mathcal{L}_{TR}$ and $\mathcal{L}_{SR}$ offer distinct levels of editing granularity, as demonstrated in Fig. 8. $\mathcal{L}_{TR}$ excels in maintaining the inherent structure within the edit region, which aids in achieving natural color and texture modifications. On the other hand, $\mathcal{L}_{SR}$ provides stronger guidance by directly amplifying the CA values within the mask, leading to more noticeable structural

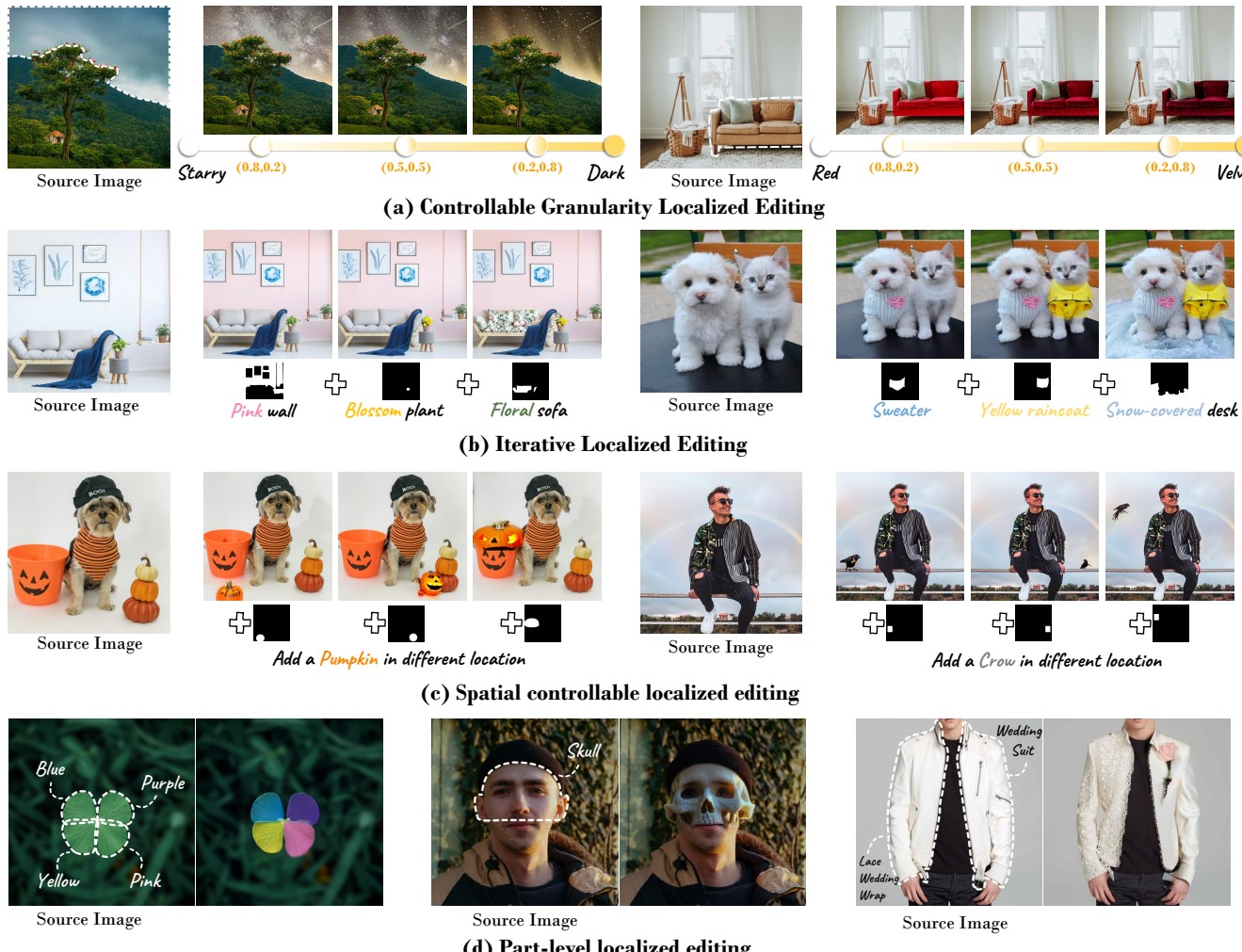

**(a) Controllable Granularity Localized Editing**

**(b) Iterative Localized Editing**

**(c) Spatial controllable localized editing**

**(d) Part-level localized editing**

**Figure 9: Other localized editing applications of the proposed MAG-Edit.**

changes in the edit region. As a result, $\mathcal{L}_{SR}$ is better suited for edits involving large structural shape changes.

## 4.4 Localized Editing Applications

As illustrated in Fig. 9, MAG-Edit proves to be effective across a range of localized editing applications. In Fig. 9(a), it demonstrates the ability to balance editing granularity for various prompts, meeting the specific needs of users for different target tokens. Moreover, Fig. 9(b) highlights MAG-Edit's capacity for performing iterative, localized modifications on multiple objects within a single image. Additionally, MAG-Edit provides control over the spatial precision of edits, as shown in Fig. 9(c). Furthermore, our method is adaptable for part-level editing, as depicted in Fig. 9(d).

## 4.5 Limitations and Future Work

MAG-Edit has demonstrated effectiveness in facilitating localized edits in complex scenarios, yet it is not without its limitations, which are pivotal to our ongoing and future research. One key limitation is the method's processing time, which is about 1.5 minutes per image on an A100 GPU, primarily due to the optimization process.

Furthermore, choosing the most suitable constraint type and deciding on the optimal number of optimization iterations are crucial factors. Additionally, our approach currently struggles with non-rigid editing in complex scenarios, a concern that will be a primary focus in our forthcoming research efforts.

## 5 CONCLUSIONS

In this work, we present a novel plug-and-play, inference-stage optimization scheme *Mask-Based Attention-Adjusted Guidance* (MAG-Edit). This method empowers attention-based editing frameworks, such as Prompt-to-Prompt (P2P), to enhance localized editing in complex scenarios that feature multiple objects and intricate compositions. In particular, we propose to maximize two mask-based CA ratios, namely the token and spatial ratio, to locally optimize the noise latent feature for enhanced alignment with the target edit token. Our experimental results, both quantitative and qualitative, consistently illustrate that MAG-Edit outperforms existing methods in localized image editing within complex scenarios. We believe that the proposed MAG-Edit scheme has pioneered a novel direction for applying localized editing in real-world scenarios.

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
