# OpenReview forum: "MAG-Edit: Localized Image Editing in Complex Scenarios via  $\underline{M}$ask-Based $\underline{A}$ttention-Adjusted $\underline{G}$uidance"
_acmmm.org/ACMMM/2024/Conference — MM2024 Poster_

### Official Review · Reviewer_sofZ · 2024-05-11

**Rating:** 4
**Confidence:** 3

**Summary:**

The paper proposes a mask-based local image editing algorithm, MAG-Edit, for complex scenes, enabling editing of target objects within complex scenes without the need for retraining. MAG-Edit optimizes latent noise features and utilizes a mask-based cross-attention mechanism to precisely align editing instructions with image details, thereby improving the accuracy and structural consistency of local edits. The problem studied in the paper has high practical value, and the proposed method is closely related to the research problem, though the theoretical innovation of the method is limited. Additionally, the authors present rich experimental results and discuss the generation results under different conditions.

**Strengths:**

1) The paper has a clear structure, with concise and understandable writing, providing a complete summary of existing methods.
2) The research problem is clearly defined and has strong practical value, with a simple and effective method.
3) The experiments are comprehensive, discussing results under various scenarios and conditions, as well as performance under different metrics.

**Limitations:**

1) The experimental results in the paper are all perfect; adding discussions of some failed cases in the appendix would provide a more comprehensive demonstration of the method's effectiveness.
2) The citation format is inconsistent; for instance, [7] uses abbreviations like ICLR while reference [10] uses the full name "International Conference on Learning Representations," similar issues exist for ICCV in [4][27] and CVPR in [2][3][25][29].
3) The paper emphasizes complex environments in the title and content, but the connection to complex environments is not apparent from the method design and experiments.
4) In the application of reference [20], there are many mask-based inpainting methods; could the authors explain the differences between their method and these methods? For example: Corneanu, Ciprian, Raghudeep Gadde, and Aleix M. Martinez. "LatentPaint: Image Inpainting in Latent Space With Diffusion Models." Proceedings of the IEEE/CVF Winter Conference on Applications of Computer Vision. 2024.
5) In the third sentence of the Abstract, the authors mention, "Existing mask-based inpainting methods fall short of retaining the underlying structure within the edit region, causing noticeable discordance with their complex surroundings." Could the authors provide specific examples? Under the same premise of using a Diffusion-based model as the base model.

**Suitability:**

3

---

### Official Review · Reviewer_6kTu · 2024-05-24

**Rating:** 4
**Confidence:** 3

**Summary:**

Recent diffusion-based image editing techniques perform well with simple images but struggle with complex images with multiple objects. Existing mask-based inpainting methods and attention-based methods like P2P have difficulties in maintaining the structure and alignment in such images. The study proposes MAG-Edit, a plug-and-play optimization method which improves localized image editing in intricate scenarios. MAG-Edit optimizes the noise latent feature to enhance the local alignment with the desired prompt. The method has proved effective in achieving text alignment and structure preservation in complex scenarios through extensive experiments.

**Strengths:**

1.It proposes two mask-based Cross Attention (CA) constraints in terms of token and spatial ratio. These are designed to maximize the CA value ratio within a predefined editing region, guiding the local noise latent feature to align better with the target editing tokens.

2.It extensively validates the efficiency of MAG-Edit in localized image editing across diverse intricate scenarios. Quantitative and qualitative experimental results demonstrate a significantly improved trade-off between editing efficiency and structure preservation compared to existing state-of-the-art approaches.

**Limitations:**

1.In my review of this paper, I noted the authors have implemented an improved attention-based mask strategy within the P2P method. This appears to be an innovative approach as this strategy aids in maintaining consistency within complex scenarios. Nevertheless, multiple similar mask strategies have been incorporated within Diffusion-based methods to achieve contiguousness in other areas, and the paper possibly does not fully justify its advantages. I would advise the authors to expound upon this aspect.

2.Regarding the formula 2 inside the "Spatial Ratio Constraint" part, I concur with your views that the formula should distinguish the mask inside and outside the area information. However, the formula multiplies the same mask M, which might lead to ambiguity. Would it be better served to replace it with (1-M)? I would recommend that the authors conduct a more comprehensive analysis and derivation in this area, and to provide their readers with additional symbol explanations.

3.Further, concerning the selection of the balance weight λ, the authors simply assign a value 3, without providing any explanation for this choice. I would suggest the authors can confirm the optimal value of λ through quantitative experiments.

4.The paper positively affirms the excellent performance of the P2P method under single object conditions. Nevertheless, the results generated by the P2P method in figure 4 (e) display unsatisfactory outcomes. For example, when the prompt word is "Limousine", no substantial change is observed in the editing results by the P2P method. Similarly, when the prompt word is "Steaks", all editing objects generated by the P2P method appear in an incorrect shade of green. I propose the authors should delve into explaining why, even under the same baseline model, there are significant differences merely considering the mask guidance.

In light of the above recommendations, I hope the authors can further refine the paper's content to help readers better understand its purpose and technical details. I also look forward to the authors' responses and clarifications to these raised concerns.

**Suitability:**

3

---

### Official Review · Reviewer_BJLP · 2024-05-26

**Rating:** 4
**Confidence:** 4

**Summary:**

The paper proposes a method for local image editing that leverages a text-to-image generative model. This approach introduces an innovative inference optimization strategy along with mask-based CA constraints to improve editing outcomes. Compared to the previous method, P2P, the proposed method significantly alleviates background inconsistencies, ensuring that the edited regions blend seamlessly with the surrounding areas. Moreover, it enhances the coherence between the edited region and the textual description provided by the user. Quantitative experiments were conducted to evaluate the method’s performance, and the results demonstrate superior performance compared to previous image editing methods. This method effectively addresses the common issues found in prior approaches, offering a more reliable and accurate solution for local image editing tasks.

**Strengths:**

1.	MAG-edit performs well compared to previous editing methods, demonstrating superior results in both qualitative and quantitative evaluations. Its advanced algorithms and robust architecture contribute to its high performance, setting a new standard in the field of image editing.
2.	MAG-edit effectively mitigates background inconsistency and editing-description inconsistency issues found in attention-based image editing methods. By employing a more refined approach, it ensures that the edited images maintain a coherent background and that the modifications align accurately with the provided descriptions, enhancing the overall quality and reliability of the edits.

**Limitations:**

1. The quantitative experimental data is insufficient, as it only includes one table and lacks comprehensive ablation experiments. This limits the ability to fully evaluate the effectiveness of the proposed method.

2. The performance of the proposed framework, both qualitative and quantitative, without the inference optimization strategy needs to be evaluated. Since the authors mentioned the issue of inference efficiency in the limitations section, assessing the performance without this strategy is crucial to understand the overall impact of the optimization.

3. The layout of Figure 4 is difficult to read. It is recommended to replace the (a)-(h) notations with clearer method names to enhance readability and comprehension.

4. Both Stable Diffusion Inpainting, SDXL Inpainting, and DALLE now support text-guided local editing. It is unclear why these methods were not included in the comparisons, given their relevance and capability for similar tasks. Including them would provide a more comprehensive and fair evaluation of the proposed framework.

**Suitability:**

3

---

### Meta-Review · Area_Chair_3Cyc · 2024-06-30

**Recommendation:** Accept (Poster)
**Confidence:** 5

**Metareview:**

The paper presents MAG-Edit, a mask-based local image editing algorithm for complex scenes. The designed method is reasonable and the experiments validates the efficiency of MAG-Edit.

The reviewers raised concerns regarding insufficient experiments, lack of comprehensive ablation experiments, insufficient explaining/discussion of P2P, and so on. After the rebuttal and discussion period, all the reviewers gave positive ratings. Given the recommendations from three reviewers, the AC recommend this paper for acceptance.